



# Carbonaceous material export from Siberian permafrost tracked across the Arctic Shelf using Raman spectroscopy

Robert B. Sparkes[1], Melissa Maher[1], Jerome Blewett[2,*], Ayça Doğrul Selver[2,3], Örjan Gustafsson[4], Igor P. Semiletov[5,6,7], and Bart E. van Dongen[2]

[1]School of Science and the Environment, Manchester Metropolitan University, Manchester, UK
[2]School of Earth and Environmental Sciences and Williamson Research Centre for Molecular Environmental Science, University of Manchester, UK
[3]Balıkesir University, Geological Engineering Department, Balıkesir, Turkey
[4]Department of Environmental Science and Analytical Chemistry (ACES) and the Bolin Centre for Climate Research, Stockholm University, Sweden
[5]Pacific Oceanological Institute Far Eastern Branch of the Russian Academy of Sciences, Russia
[6]International Arctic Research Center, University of Alaska, USA
[7]National Tomsk Research Polytechnic University, Russia
[*]Now at: Organic Geochemistry Unit, School of Chemistry, Cabot Institute, University of Bristol, Bristol, UK

*Correspondence to:* Robert Sparkes (r.sparkes@mmu.ac.uk)

**Abstract.** Warming-induced erosion of permafrost from Eastern Siberia mobilises large amounts of organic carbon and delivers it to the East Siberian Arctic Shelf (ESAS). In this study Raman spectroscopy of Carbonaceous Material (CM) was used to characterise, identify and track the most recalcitrant fraction of the organic load. 1463 spectra were obtained from surface sediments collected across the ESAS and automatically analysed for their Raman peaks. Spectra were classified by their peak

5   areas and widths into Disordered, Intermediate, Mildly Graphitised and Highly Graphitised groups, and the distribution of these classes was investigated across the shelf. Disordered CM was most prevalent in a permafrost core from Kurungnakh Island, and from areas known to have high rates of coastal erosion. Sediments from outflows of the Indigirka and Kolyma rivers were generally enriched in Intermediate CM. These different sediment sources were identified and distinguished along an E-W transect using their Raman spectra, showing that sediment is not homogenised on the ESAS. Distal samples, from the

10  ESAS slope, contained greater amounts of Highly Graphitised CM compared to the rest of the shelf, attributable to degradation or, more likely, winnowing processes offshore. The presence of all four spectral classes in distal sediments demonstrates that CM degrades much slower than lipid biomarkers and other traditional tracers of terrestrial organic matter, and shows that alongside degradation of the more labile organic matter component there is also conservative transport of carbon across the shelf toward the deep ocean. Thus, carbon cycle calculations must consider the nature as well as the amount of carbon liberated

15  from thawing permafrost and other erosional settings.

## 1 Introduction

Extensive northern hemisphere permafrost deposits contain approximately 40 % of the global soil organic carbon (OC) budget (Schuur et al., 2015; Tarnocai et al., 2009). The majority of this carbon is currently freeze-locked, but rapid warming at high





latitudes is making it increasingly vulnerable to mobilisation via fluvial and coastal erosion. Thawing also leads to degrada-tion, both in-situ and post-mobilisation, to greenhouse gases, providing a positive feedback response to warming (Stendel and Christensen, 2002; Vonk et al., 2010, 2015). The Eurasian Arctic region contains the majority of northern hemisphere per-mafrost OC, and the release rate of both OC and sediment from this area into the Arctic Ocean is predicted to rise during the

next century (van Dongen et al., 2008; Holmes et al., 2002, 2012; O'Donnell et al., 2014; Peterson et al., 2002) Deepening hydrological flow paths, and thermokarst erosion events, are mobilising 'old' carbon that has been stored in deep permafrost for thousands of years (Gustafsson et al., 2011; Feng et al., 2013, 2015; Vonk et al., 2012; Vonk and Gustafsson, 2013; IPCC, 2013). OC delivered by fluvial erosion and transport has been identified offshore using molecular biomarkers, and shown to deposit and/or degrade rapidly on the shelf (Bröder et al., 2016; Doğrul Selver et al., 2015; Karlsson et al., 2015; Sparkes et al.,

2015, 2016; Tesi et al., 2014, 2016). In addition to fluvial erosion, coastal erosion delivers a significant amount of sediment and OC ($44 \pm 10$ TgOCy$^{-1}$) to the Arctic Ocean (Vonk et al., 2012). Poorly lithified coastal cliffs, combined with stormy weather, have caused erosion rates of up to 10 m yr$^{-1}$ (Lantuit et al., 2011). Reducing sea ice cover, and therefore increased wave fetch and storm impact, is also likely to increase coastal erosion rates during the next century (Feng et al., 2015; Stein and MacDonald, 2004; Vonk et al., 2010).

## 1.1 Recalcitrant organic matter export from permafrost

Erosion of ancient carbon from Arctic permafrost may lead to degradation of carbon into greenhouse gases and a positive feedback effect on global climate, or the absence of degradation may allow this material to be used to track eroded sediment across the continental shelf. Alongside modern OC transport, erosion mobilises petrogenic OC from rocks and soils (also known as carbonaceous material, CM) and delivers this to the Ocean. CM consists of recalcitrant material, such as coal, lignite,

combustion-derived black carbon, kerogen (insoluble hydrocarbons; Durand, 1980) and graphite, which is stable over millions of years. CM is (apart from black carbon) formed from the maturation of buried organic matter, with the degree of diagenesis and metamorphism controlling the molecular and crystalline structure (Beyssac et al., 2002b, 2003a, 2007). High temperatures during burial and metamorphism drive the transition from disordered kerogen and lignite to highly crystalline graphite, allowing CM from different areas to be identified due to differences in the geological history of the source rocks (Sparkes et al., 2013).

CM represents a major fraction of the global OC budget, sequestered in sedimentary and metamorphic rocks over million-year timescales (Bouchez et al., 2010; Galy et al., 2007; Hilton, 2008; Smith et al., 2013). Any oxidation of CM during erosion, transport and burial represents a movement of carbon between the long-term and short-term carbon cycles. During transport, bioavailability of CM is believed to be low, but there is evidence that extremely long transport distances can degrade disordered CM, leaving only crystalline graphite (Bouchez et al., 2010; Galy et al., 2007). Previous work has investigated CM in tropical

and sub-tropical settings, but, in contrast to extractable OC, no work has been done to characterise or trace CM in the Arctic. The only studies on recalcitrant Arctic OC involve (usually submicron-sized) soot black carbon particles (Coppola et al., 2014; Salvadó et al., 2017; Winiger et al., 2017).

Salvadó et al. (2017) measured the distribution of soot black carbon (SBC) across the ESAS. SBC was isolated using chemothermal oxidation at 375 °C, a method that has been throughly tested to isolate combustion-derived SBC from other



recalcitrant species (such as coal of various degree of maturity, pollen and other biomacromolecues; Gustafsson et al., 2001; Elmquist et al., 2006). The highest concentrations (up to 0.22 wt%SBC) and proportions (11% of the Total OC) of SBC were found at the mouths of the Lena and Kolyma Rivers, with concentrations and proportions decreasing offshore as SBC was deposited and marine primary productivity became the dominant OC source. However, the relative importance of the

recalcitrant SBC within the permafrost-derived carbon pool increased as more labile material degraded during transport. This pattern was seen consistently along a W-E transect from the Laptev Sea to the Eastern ESAS. The authors concluded that the source of SBC was not atmospheric deposition (e.g. via biomass burning) but permafrost erosion. Goñi et al. (2005) report anomalously old radiocarbon ages for distal sediments on the Beaufort Shelf, attributed to highly recalcitrant terrestrial OC that has been matured to kerogen grade or higher, but the nature and origin of the material has not been investigated in detail.

Therefore, there is a need to assess whether the various inputs of carbon to the Arctic Ocean system can explain these old radiocarbon ages, and whether they form an active or passive part of the carbon cycle.

This study uses Raman spectroscopy to identify the sources of CM in the East Siberian region, and tracks the export of CM from these sources across the East Siberian Arctic Shelf (ESAS). This is the first Raman study of sedimentary CM in the Arctic. Automated Raman spectra fitting procedures were used to analyse > 1400 spectra measured from CM in sediments

collected across 1 million $km^2$ of the (ESAS). Multiple heterogeneous sources of sediment were studied, including several of the world's largest river catchments and thousands of km of shoreline experiencing rapid coastal erosion (Lantuit et al., 2011).

## 1.2  Raman Spectroscopy

Raman spectroscopy is a precise and powerful tool for quantifying the degree of crystallinity within CM (Beyssac et al., 2002a, 2003b). In this study, it is used to identify permafrost-sourced CM offshore. Monochromatic incident light interacts with the

crystal lattice of the targeted particle, and reflects with a changed wavelength due to the energy shift introduced by lattice vibrations (phonons). When analysed with in a Raman spectrometer, graphitic and disordered carbon can be characterised by studying the response within the range 800 – 2000 wavenumbers. In a pure graphite crystal, lattice vibrations are restricted to bond stretching of $sp^2$ atom pairs only, leading to only one Raman peak at 1580 $cm^{-1}$ (known as G1). Disordered CM allows for 'breathing' or aromatic rings, and longitudinal oscillations (Ferrari, 2007). This introduces more peaks into a Raman

spectrum, at 1350 cm−1 (D1), 1620 cm−1 (D2), 1500 cm−1 (D3) and  1200 cm−1 (D4). These peaks appear and grow as the degree of disorder increases. Highly graphitised CM is dominated by the G peak, with minor contributions from D1 and D2. In a highly disordered sample the largest peak will be D1, the D2 and G peaks will overlap to form a single peak at approximately 1600 cm−1, and the D3 and D4 peaks will form a noticeable part of the spectrum (see Supplementary Figure 1). Advances in computing power and have allowed the analysis of these spectra to transition from manual peak fitting

to an automated approach, in which a combination of difference-minimising algorithms and defined limits fits multiple peaks to a Raman spectrum. CM ranging from highly disordered to highly graphitised can be characterised and differentiated (Lee et al., 2016; Sparkes et al., 2013). Further details of the fitting script are found in section 2.4.

Since Raman spectroscopy can differentiate CM in lithologies that have experienced different metamorphic conditions, it has been used as a geothermometer in orogenic belts (Beyssac et al., 2007). Peak area ratios have been calibrated against





metamorphic temperatures for mildly to highly graphitised CM (Beyssac et al., 2002a, 2003b), and for more disordered CM (Lahfid et al., 2010). Furthermore, the recalcitrance of CM allows it to be identified downstream following weathering and erosion. CM, especially highly graphitised material, eroded from the Himalaya and Andes has been identified thousands of kilometres downstream in the Bengal Fan and Amazon River (Galy et al., 2007; Bouchez et al., 2010). During long distance

transport, disordered CM appears to be preferentially degraded. Over shorter distances, the relative distribution of disordered and graphitised CM allows the use of Raman spectroscopy as a tool for tracing erosion from separate lithologies. This technique has been applied in single river catchments in Taiwan and New Zealand (Nibourel et al., 2015; Sparkes et al., 2013) but up to now has not been applied to wider, continental shelf systems. Given the complex interplay between fluvial and coastal erosion in the East Siberian region, this is the most geologically complex application of Raman spectroscopy of CM to date. Raman

spectroscopy was used to characterise the CM in each sediment sample (disordered through to extremely graphitised), with the changing distribution of spectral groups being used to: (i) identify whether extremely heterogeneous sedimentary systems can be characterised using Raman spectroscopy of CM; (ii) differentiate the CM present in sediments delivered to the ESAS via fluvial and/or coastal erosion; (iii) track the CM (and therefore sediment) from these various inputs across the ESAS; and (iv) evaluate the impact of CM erosion and transport on the carbon cycle in Arctic permafrost systems

## 2  Materials and Methods

### 2.1  Study Area

Samples used in this study were collected from across the East Siberian Arctic Shelf, from 130 to 175 °E and from 70 to 77 °N. This area includes the outflows of four of the major rivers; the Lena, Yana, Indigirka and Kolyma Rivers (see Figure 1). Collectively, these rivers drain 3680 $\times 10^3$ km$^2$ of Siberia, including tundra and taiga environments. This area is largely

underlain by permafrost, where soil temperatures remain below 0 °C year-round. Northern and eastern areas contain continuous permafrost, where frozen ground forms a layer impenetrable to water, whilst discontinuous permafrost in the southern and eastern parts of the catchment allows water ingress to lower soil levels (Kotlyakov and Khromova, 2002). The uppermost portion (also known as the "active layer") freezes and thaws on an annual basis and contain both open tundra and taiga forests. The active layer is expected to deepen as the climate warms in the next century, which will also lead to a reduction in continuous

permafrost area and an increase in fluvial erosion (Feng et al., 2015). Additionally, the East Siberian coastline contains large Ice Complex Deposits (ICD; also known as "Yedoma"). These Plio-Pleistocene permafrost deposits are weakly lithified and rich in well-preserved OC, providing a major influx of sediment and carbon to the Arctic Ocean (Bischoff et al., 2016; Lantuit et al., 2013; Schirrmeister et al., 2008, 2011; Strauss et al., 2012, 2013; Sparkes et al., 2016; Vonk et al., 2010, 2012).

    Sediment sourcing in Eastern Siberia is due to fluvial and coastal erosion. Runoff from the surface carries soil from the active

layer into lakes and rivers. Lateral cutting can mobilised sediment from deep permafrost layers directly into the rivers. The Lena, Yana, Indigirka and Kolyma rivers deliver 523, 32, 54 and 122 km$^3$ y$^{-1}$ of water and 21, 4, 11 and 10 Mt y$^{-1}$ sediment respectively (Gordeev, 2006). Coastal erosion is another major source of sediment to the ESAS. Coastal erosion rates of up to



$10 \, \mathrm{m} \, \mathrm{y}^{-1}$ have been measured (see Figure 1), among the fastest in the Arctic (Lantuit et al., 2011). Erosion rates from ICD are five to seven times greater than other coastal permafrost (Lantuit et al., 2011).

The river catchments in this region extend as far south as 52 °N, underlain by a wide range of lithologies. Carbonaceous material exported to the ESAS could have been transported over 4000 km from the Lena River headwaters, over decadal to
millennial timescales if floodplain deposition is taken into account. This is a much larger system that has been studied using the Raman sedimentary provenance tool before (Sparkes et al., 2013; Nibourel et al., 2015). Potential CM source lithologies include extensive coal basins within the Lena catchment (Kuznetsov et al., 2009), and metamorphic zones within the Chersky collision belt that have experienced Greenschist to Amphibolite metamorphic conditions, with temperatures up to 620 °C (Oxman, 2003). This is enough to form crystalline graphite (Beyssac et al., 2002b).
The ESAS is extremely shallow, generally less than 100 m (see Figure 1). The seabed consists of permafrost that developed sub-aerially, was flooded during Holocene sea-level rise and is now being buried by sediment sourced from fluvial and coastal erosion(Kienast et al., 2005). Geochemical studies investigating the sources and offshore distribution of organic matter have noted differences between east and west, nearshore and offshore sections of the shelf (Bischoff et al., 2016; Karlsson et al., 2015; Semiletov et al., 2005; Sparkes et al., 2015, 2016; Tesi et al., 2014).

## 2.2 Sample Collection

This study analysed 56 samples collected from terrestrial and marine settings in Eastern Siberia. 47 were surface sediment samples (samples YS-XX and TB-XX) collected from across the ESAS during the International Siberian Shelf Study research cruise in 2008 (ISSS-08; Semiletov and Gustafsson, 2009). GEMAX sediment cores were sliced into 1 cm sections and transferred into pre-cleaned polyethylene containers, van Veen grab samples were sub-sampled using stainless steel instruments
into pre-cleaned polyethylene containers. Only the top 0-1 cm of cores, and the upper 2 cm of grab samples, were used in this study. Three terrestrial locations provided information on permafrost material from the Lena, Indigirka and Kolyma river catchments. In the Lena Delta, this comprised subsamples from a permafrost core collected from Kurungnakh Island (samples KUR-XXXX; Russian-German LENA 2002 expedition; Bischoff et al., 2013; Grigoriev et al., 2003). Subsamples at 0.34 m, 14.4 m and 22.0 m were analysed. Six ICD samples were collected from the upper, middle and lower portions of riverbank
profiles in the Indigirka River (samples KY-XXX) and Kolyma River (samples CH-XXX) catchments (Tesi et al., 2014). All samples were kept frozen below -18°C before being stabilised by freeze-drying. Sample locations are shown in Figure 1 and reported in the "Sample Metadata" supplementary table.

A minority of sediments used in this study were solvent extracted for biomarker analysis prior to collecting Raman spectroscopy measurements (Bischoff et al., 2016; Doğrul Selver et al., 2015; Sparkes et al., 2015). An ultrasonic extraction process
used methanol, dichloromethane and pH-buffered distilled water in order to remove extractable material. This represents approximately 5 % of the total OC content, OC that is not likely to be Raman-amenable. Comparison of Raman spectra collected from extracted and non-extracted sediments showed that there was no noticeable effect on the distribution of spectra; for completeness the extracted sediments have been identified in the "Sample Metadata" supplementary table.





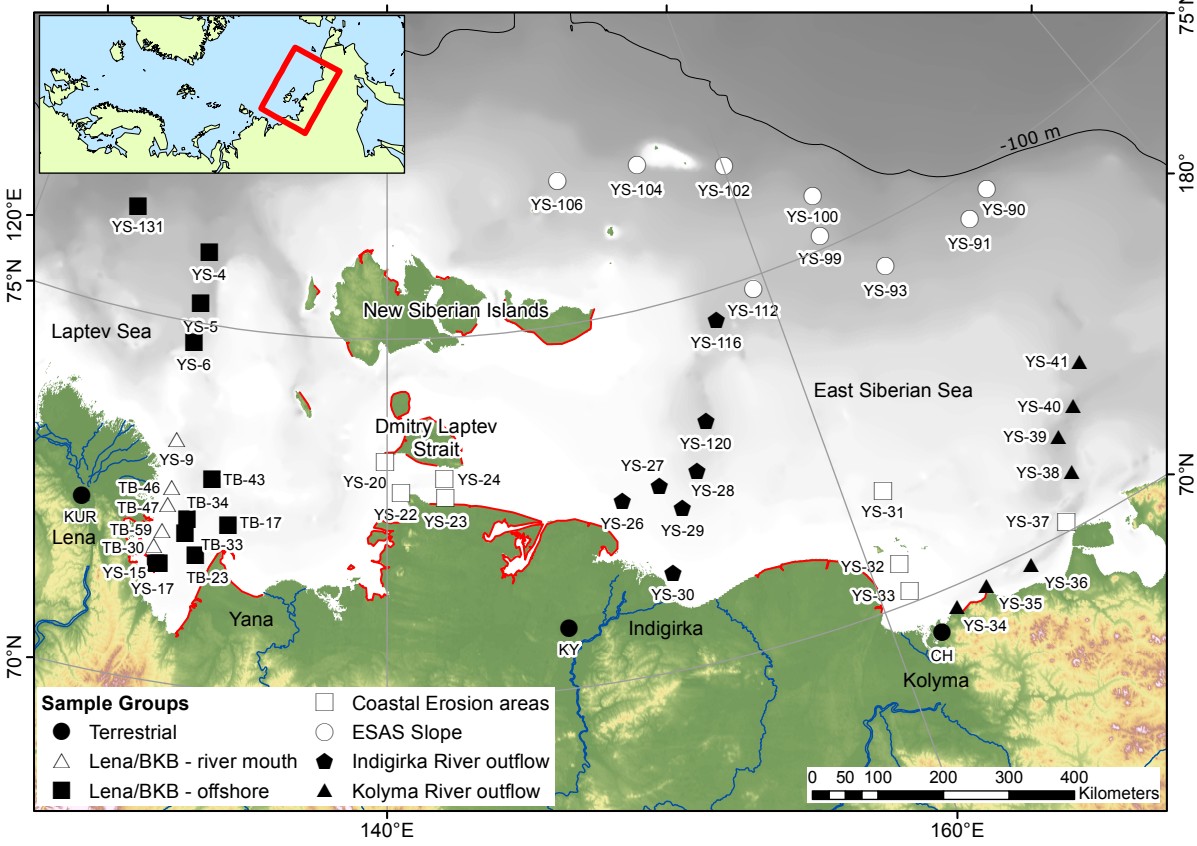

**Figure 1.** Map of the East Siberian Arctic Shelf (ESAS) showing the location of surface sediment and permafrost samples used in this study. Thin black line represents the -100 m bathymetric contour. Grey to white colours offshore represent the transition from -100 m to 0 m water depth, highlighting both the shallow nature of the ESAS and the minor bathymetric features present. Major rivers are labelled. Areas of rapid coastal erosion (>1 m y$^{-1}$; Lantuit et al., 2011) are shown in red. Groups of samples are denoted by marker shape and colour: black circles = terrestrial ICD and permafrost cores; white triangles = nearshore Lena River outflow / Buor-Khaya Bay; black squares = offshore Lena River outflow / Buor-Khaya Bay; white squares = areas of high coastal erosion and little fluvial input; white circles = distal ESAS areas; black pentagons = Indigirka River outflow; black triangles = Kolyma River outflow.

## 2.3 Raman Spectroscopy

Raman spectra were collected following the procedure of (Sparkes et al., 2013). Approximately 1 g of sediment from each sample was collected and homogenised by stirring in a glass vial. Samples with sand or silt sized grains were crushed using a pre-cleaned pestle and mortar to obtain a fine-grained sediment. It has been shown that physical grinding, even over several hours, does not affect carbonaceous material crystallinity (Nakamizo et al., 1978; Sparkes et al., 2013). Core samples (KUR-XXXX) were ground using a planetary mill. A subsample of 0.25 g was taken and placed onto a glass slide, forming a circle



of diameter 1-2 cm. This was pressed below another slide to produce a uniform horizontal surface in order to aid focussing and align tabulate particles with the incident light beam. The sample was rastered under the microscope and all potential CM particles were investigated using a brief exposure (2 s at 10 % laser power). Those confirmed as being carbonaceous were analysed using an extended acquisition, using the following settings. Samples exposed to a 514 nm Argon ion laser set at

~1 W power, under a 50x magnifying lens. The reflected light was analysed using a Renishaw InVia Raman spectrometer with a grating spacing of 1800 $\mathrm{lmm}^{-1}$. Synchroscan mode was used to allow a spectral window of 800 to 2200 $\mathrm{cm}^{-1}$. Three 20 s acquisitions were made at 10 % laser power for a total of 60 s acquisition time. Where possible, thirty spectra were collected from each sample. Data was exported as a text file (wavenumber, intensity) for subsequent analysis.

## 2.4  Automated processing of Raman spectra

Raman spectra from carbonaceous material are often complex, consisting of up to five overlapping peaks with an additional offset increasing with wavenumber (see Supplementary Figure 1). The peak width, and number of peaks, is highest in the most disordered CM (Beyssac et al., 2002a, 2003b; Lahfid et al., 2010; Sparkes et al., 2013). An automated computer script was used to quantify, quickly and objectively, the peak locations, heights, widths and areas for further analysis (Sparkes et al., 2013). Briefly, the script determines whether it is appropriate to use three Voigt peaks (Beyssac et al., 2002b, 2003a)

or five Lorentzian peaks (Lahfid et al., 2010) to best fit the spectrum. These peaks are adjusted to find the optimum combination of peak location, width and height using the best-fit algorithms implemented by the open source software package Gnuplot (version 4.6). A linear background subtraction was made as part of this automated fitting. For full details of the fitting process, see Sparkes et al. (2013). The fitting script used in this study is included as a Supporting File, and developed at https://github.com/robertsparkes/raman-fitting.

As well as peak shape parameters (location, height, width, area), the fitting script calculates two additional values. The implied maximum burial temperature experienced by the carbonaceous material is calculated based on the peak area ratio. For mildly and highly graphitised material, fitted using three Voigt peaks, the R2 ratio compares the G, D1 and D2 peaks (Beyssac et al., 2002b):

$$R2 = \frac{D1}{G + D1 + D2} \tag{1}$$

For more disordered CM fitted using five Lorentzian peaks, the RA2 ratio compares the G, D1, D2, D3 and D4 Lorentzian peaks (Lahfid et al., 2010):

$$RA2 = \frac{(D1 + D4)}{(G + D2 + D3)} \tag{2}$$

The temperature value derived from these calculations represents the degree of graphite crystallinity in the carbonaceous material. Highly graphitic samples have higher metamorphic temperatures, up to 650 °C (Beyssac et al., 2002a). Less graphitised

samples have lower temperatures, calibrated down to 200 °C (Lahfid et al., 2010). It relates to the peak metamorphic conditions experienced by the CM (Beyssac et al., 2002b), and when dealing with sedimentary samples is used as a tracer for CM from different lithologies (Sparkes et al., 2013). The second parameter is the "sum of peak widths" parameter, which is the sum of





**Table 1.** Parameters for classifying Raman spectra into four groups based on their R2 or RA2 peak area ratio, and Totalwidth parameter (G + D1 + D2). Note that there are overlaps between the defined regions. Mildly Graphitised takes precedence over Intermediate grade, if a sample plots in the overlapping region.

| Spectral group | $T_{min}$ | $T_{max}$ | $Width_{min}$ | $Width_{max}$ |
| --- | --- | --- | --- | --- |
| | °C | °C | cm$^{-1}$ | cm$^{-1}$ |
| Disordered | 138 | 384 | 290 | 400 |
| Intermediate | 138 | 384 | 80 | 290 |
| Mildly Graphitised | 350 | 500 | 40 | 250 |
| Highly Graphitised | 500 | 650 | 40 | 250 |

the G, D1 and D2 peak widths at half-maximum. This value is lowest for highly graphitised CM, and increases with increasing CM disorder. Although it is a non-standard method for classifying Raman spectra, this parameter differentiates CM better than other potential parameters. For example, measuring just the width of peak D1 can limit the ability to differentiate between moderately and extremely disordered CM, as the D1 peak width saturates at high amounts of disorder. When analysing only mildly and highly graphitised CM, the D1 width parameter could be used instead of total width. Extremely disordered CM has the highest total width value, and this factor can be used to discriminate between samples that are mildly metamorphosed (and legitimately sit on the calibration of Lahfid et al. (2010)) and those that have undergone only diagenetic alteration (e.g. lignite-grade CM). These extremely low-grade CM particles, for which metamorphic temperature calibrations are not available, can still be distinguished and identified via their Raman spectra despite the lack of calibrated temperature (Sparkes et al., 2013). Using a combination of these two parameters, spectra were divided into four classes: Disordered; Intermediate; Mildly Graphitised; and Highly Graphitised (see Table 1).

## 3 Results

### 3.1 Raman spectra collection

The 1463 spectra collected span the entire range from perfectly crystalline graphite to highly disordered CM. Maximum implied metamorphic temperatures were 641 °C , i.e. spectra with no discernible D1 peak. Highest total peak width values were 366 cm$^{-1}$, i.e. extremely disordered spectra implying minimal CM crystallinity. Between these two endmembers, a complete range of spectra was collected. When rastering the microscope focus point across the samples, CM was easiest to find and measure in nearshore samples. CM in samples collected from the distal shelf was more disperse, and often consisted of smaller particles. However, since samples were ground before analysis, the size of each CM particle was not recorded.





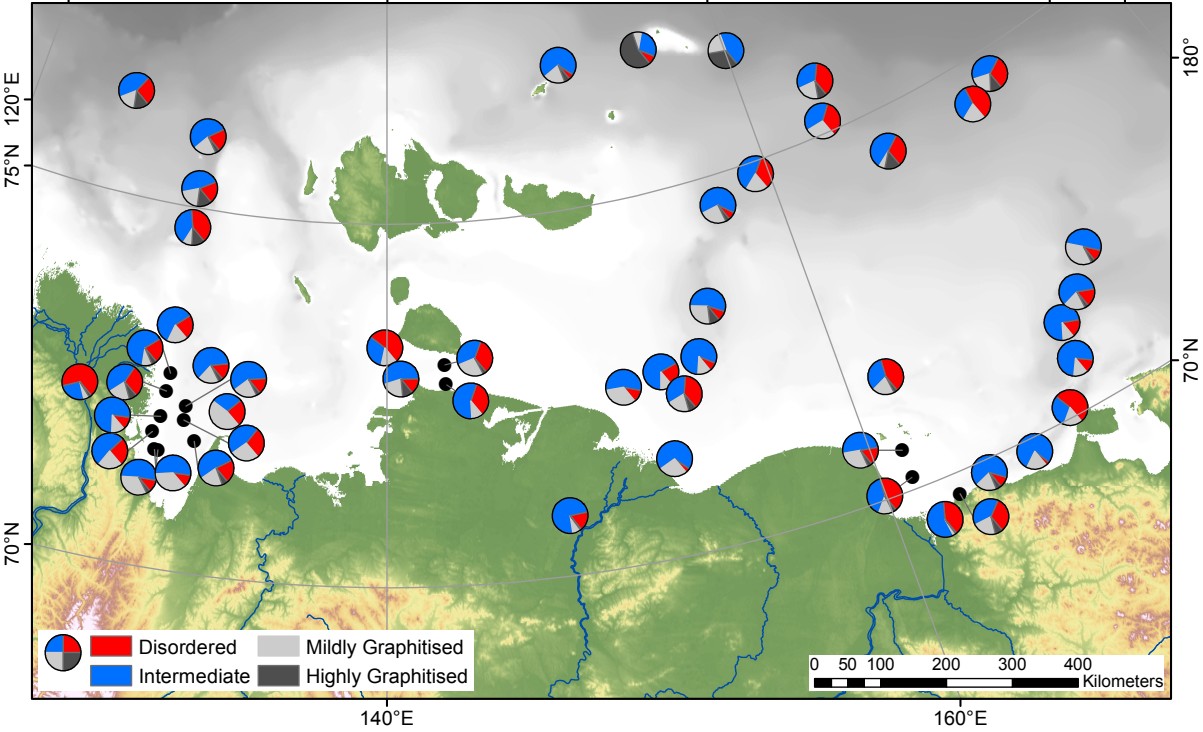

**Figure 2.** Raman spectroscopy results showing the relative amount of each carbonaceous material spectral class within each sample as pie charts. Red = Disordered CM; blue = Intermediate CM; light grey = Mildly Graphitised CM; dark grey = Highly Graphitised CM.

## 3.2 Grouping spectra by Raman parameters

For each sample, Raman spectra were classified into Disordered, Intermediate, Mildly Graphitised and Highly Graphitised CM as described above. This allowed the proportion of each spectral type in each sample to be investigated statistically. The most common group of spectra was Intermediate, comprising 50 % of the dataset. 27 % of the spectra were classified as Disordered, 18 % as Mildly Graphitised, and 5 % as Highly Graphitised. Based on the location of samples on the shelf, and guided by the proportion of spectra in each group, groupings were defined for various regions of the ESAS (Figure 2) as follows:

1. Terrestrial samples, comprising the Kurungnakh Island permafrost core, and permafrost samples collected from the Indigirka and Kolyma catchments.

2. Buor-Khaya Bay samples collected within 55 km of outflows from the Lena River (as measured by ArcGIS – see values in the "Sample Metadata" supplementary table).

3. Buor-Khaya Bay samples collected further than 55 km from outflows of the Lena River.

4. Indigirka River outflow samples.

5. Kolyma River outflow samples.

6. Samples collected from areas known to have high rates of coastal erosion (Lantuit et al., 2011). This includes the Dmitry



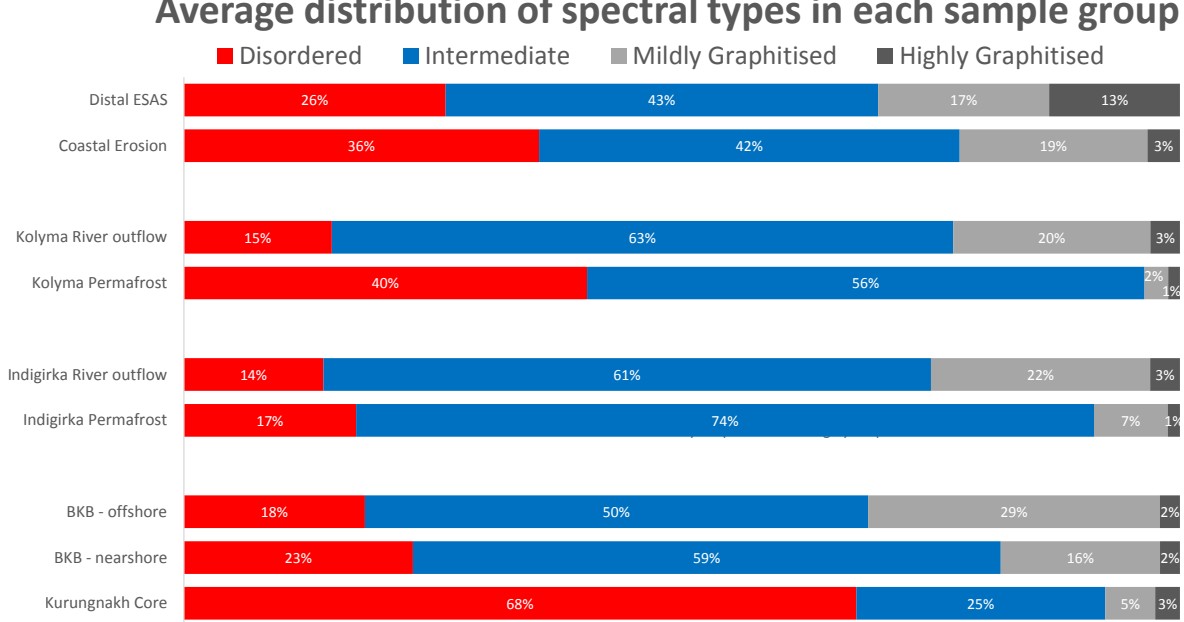

**Figure 3.** Bar chart showing the distribution of each spectral class within each group of samples. Sample groups are mapped in Figure 1 and listed in the "Sample Metadata" supplementary table. Red = Disordered CM; blue = Intermediate CM; light grey = Mildly Graphitised CM; dark grey = Highly Graphitised CM.

Laptev Strait, samples YS-31, YS-32 and YS-33, which lie to the west of the Kolyma River, and sample YS-37 which lies near Ayon Island. According to Lantuit et al. (2011), Ayon Island is eroding slowly (0-1 m yr$^{-1}$) compared to the Dmitry Laptev Strait and west-of-Kolyma regions (2-10 m yr$^{-1}$). However, sample YS-37 was included in the coastal erosion group since the proximity of sample YS-37 to the island (20 km) is significantly lower than its proximity to the Kolyma River (245 km), and the spectral distribution in sample YS-37 is significantly different if compared to neighbouring samples in the Kolyma region (YS-34 to YS-41; see Figure 2).

7. Distal ESAS samples, especially from the start of the continental slope. The ESAS is particularly shallow for hundreds of kilometres offshore, and these samples were collected in water depths ranging from 32 to 69 m.

Within each of these sample groups, the distribution of spectra is shown in Figure 3 and Table 2.

## 3.3 Distribution of spectral classes within each group of samples

The terrestrial group contains the largest proportion of Disordered CM (42 %) and the smallest proportion of Mildly Graphitised (5 %) and Highly Graphitised CM (2 %). The Buor-Khaya Bay nearshore and offshore samples contain much less Disordered CM than the terrestrial Kurungnakh Island core samples, (23 % and 18 % for the BKB vs 68 % for the Kurungnakh Island core





**Table 2.** Distribution of spectral classes within each group of samples. Percentage values are averages of the proportions within each group, rather than the proportion of the entire group. This corrects for samples in which it was not possible to collect 30 Raman spectra.

| Sample Group | $n_{samples}$ | $n_{spectra}$ | Disordered | Intermediate | Mildly Graphitised | Highly Graphitised |
|---|---|---|---|---|---|---|
| Terrestrial | 9 | 245 | 42% ± 21% | 52% ± 20% | 5% ± 2% | 2% ± 1% |
| BKB – nearshore | 5 | 132 | 23% ± 6% | 59% ± 11% | 16% ± 5% | 2% ± 3% |
| BKB – offshore | 7 | 197 | 18% ± 6% | 50% ± 10% | 29% ± 9% | 2% ± 2% |
| Indigirka outflow | 7 | 134 | 14% ± 11% | 61% ± 13% | 22% ± 8% | 3% ± 4% |
| Kolyma outflow | 7 | 188 | 15% ± 8% | 63% ± 13% | 20% ± 8% | 3% ± 3% |
| Coastal Erosion | 8 | 244 | 36% ± 14% | 42% ± 10% | 19% ± 6% | 3% ± 3% |
| Distal ESAS | 13 | 325 | 26% ± 14% | 43% ± 10% | 17% ± 7% | 13% ± 15% |

samples), and contains an increasing amount of Mildly Graphitised CM offshore (16 % and 29 %). Indigirka River and Kolyma River outflow samples contain the highest amounts of Intermediate CM (61 % and 63 % respectively) and the lowest proportion of Disordered CM (14 % and 15 %). In contrast, coastal erosion samples contain relatively high proportions of Disordered CM (36 %) and the joint-lowest amount of Intermediate CM (42 %). Distal ESAS samples contain 43 % Intermediate CM, and the highest amount of Highly Graphitised CM (13 %). The highest amount of Highly Graphitised CM was found in the two samples positioned furthest from the continental mainland, excluding the New Siberian Islands, (YS-104 (56 %) and YS-102 (33 %); see Figure 2).

## 4 Discussion

### 4.1 Distribution of Highly Graphitised CM

The relative proportion of Highly Graphitised CM in the Distal ESAS sample group was much higher than the other samples (see Table 2 and Figure 3). Throughout the Distal ESAS group, Highly Graphitised CM represents 12 % of all spectra collected (38 spectra out of 325). Averaging the proportion of Highly Graphitised in each of these samples, the Distal ESAS group has 13 % ± 15 % (1 s.d.) Highly Graphitised CM, with a range from 0 % to 55 %. This heterogeneity is seen in Figure 2, but there is still significantly (P=0.0001) more Highly Graphitised CM in the Distal ESAS group than the remaining samples (3 % ± 3 %), and the greatest proportion of Highly Graphitised CM found in any other sample is 10 % (sample YS-22). Two samples, YS-102 and YS-104, have extremely high amounts of Highly Graphitised CM, far exceeding the proportion measured in any other sample (33 % and 56 % respectively). Even without these samples in the calculation, the Distal ESAS group is still significantly different from the remaining samples (7.4 % ± 5.5 % c.f. 3 % ± 3 %, P=0.0001).



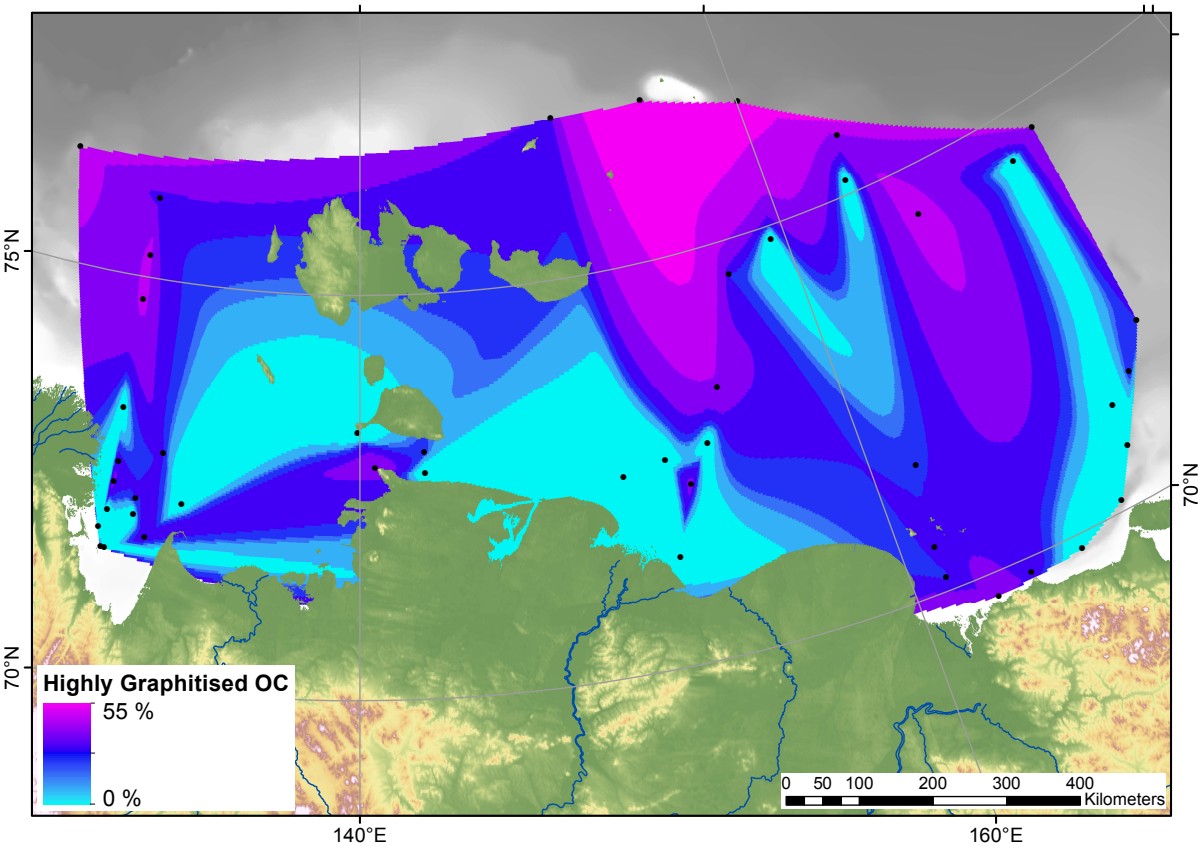

**Figure 4.** Map showing the distribution of Highly Graphitised CM across the ESAS. Distribution was interpolated using the Nearest Neighbour function within ArcMap. Colour scale transition has a geometric profile to enable regions with smaller amounts of Highly Graphitised CM to be differentiated.

There is more Highly Graphitised CM in the Distal ESAS samples than in both the nearshore ESAS samples and the terrestrial samples (1.7 % $\pm$ 1.9 % for terrestrial samples c.f. 7.4 % $\pm$ 5.5 % in the distal ESAS; P<0.05). There is an increased amount of Highly Graphitised CM in the nearshore ESAS than the terrestrial samples, but this increase is not significant (2.7 % $\pm$ 2.9 % for the non-slope offshore samples, 1.7 % $\pm$ 1.9 % for terrestrial samples; P=0.19). Therefore, there must be a process

5 leading to an enhanced amount of Highly Graphitised CM present in the furthest offshore samples. Three possibilities exist for this: unseen sources of graphite, preferential preservation of graphite, and sorting of sediment particles.

The terrestrial samples available to this project may not represent the entire material eroded to the ESAS. Eastern Siberia contains some of the world's largest drainage basins and longest rivers, and therefore CM could be eroded from thousands of kilometres inland before being delivered to the ESAS. Highly Graphitised CM has been shown to survive long-distance river

10 transport (Bouchez et al., 2010; Galy et al., 2007). There could be a source of graphite inland that, coupled with sorting effects, is responsible for the offshore graphite. In this case, rivers would deliver Highly Graphitised CM that is diluted by erosion of





non-graphitic CM close to the river mouth and along the coastline. Further processing offshore, either the loss of non-graphitic CM or the concentration of graphitic CM, means that while the graphite is a minor component of the nearshore samples, it can be seen in the distal samples.

However, samples collected near to the Lena and Indigirka river mouths show no increase in graphite compared to nearby

areas dominated by coastal erosion, suggesting that there is not even a diluted graphite signature coming from the rivers. Nearshore BKB samples (2.3 % ± 2.9 %) and the sample nearest to the Indigirka River (YS-30; 0 %) have little or no Highly Graphitised CM. The sample closest to the Kolyma River outflow has some Highly Graphitised CM (YS-34; 7 %), so there may be graphite erosion from within the Kolyma catchment. Nokleberg et al. (1993) report graphite bearing ore bodies from multiple locations within the Kolyma catchment, which form part of the Kolyma-Omolon superterrane metamorphic belt. This

material could survive transport through the Kolyma River (Bouchez et al., 2010), explaining the increase in Highly Graphitised CM see in the north eastern part of the ESAS. Given the increasing proportion of Highly Graphitised CM observed offshore across the entire E-W transect, it is more likely that the graphite is sourced in low concentrations across the region and that offshore processes are responsible for increasing the proportion in distal areas.

Regardless of the graphite source, other processes are required to explain the offshore trend in Highly Graphitised CM. There

is a clear offshore transition from lower to greater amounts of Highly Graphitised CM (Figure 4), which could have been by a preservation effect. As seen in transects along the Bengal Fan, there is often a trend to more crystalline graphite with transport distance (Galy et al., 2007). This has been explained as a resistance to physical, chemical and/or biological degradation. If Disordered, Intermediate or Mildly Graphitised CM is preferentially degraded during transport across the ESAS, distal samples would be relatively enriched in Highly Graphtised CM even if the proportion of this material in the sediment delivered to the

shelf is low. However, this pattern is not seen in the other classes of spectra. If graphitisation increases resilience to degradation, it would be expected that Disordered CM was the least resistant group of CM, and that this would be seen as a trend away from Disordered CM offshore. However, this is not the case and the Distal ESAS samples contain large amounts of Disordered CM. All CM analysed in this study must be autochthonous, even Disordered CM. In-situ production or flocculation of marine organic matter could not produce Raman-amenable CM particles.

There could also be a sorting effect across the shelf, rather than degradation. Whereas solvent extractable biomarkers of terrestrial processes have been shown to reduce rapidly across the ESAS (Bröder et al., 2016; Bischoff et al., 2016; Sparkes et al., 2015), most of the Raman-amenable CM present in these samples is likely to be resistant to degradation (Galy et al., 2008). Therefore, an alternative reason for Highly Graphitic CM to be present in high concentrations on the Distal ESAS is if it is preferentially delivered and deposited there. Tesi et al. (2016) showed that organic carbon was mostly associated with

fine particles in the distal ESAS, and that larger particulate OC was deposited close to the coastline. Graphite flakes have a lower density than many silicate minerals, and have a tabulate form. Winnowing is used commercially to separate graphite flakes from bulk sediments using liquid froth or air jet (Mitchell, 1993). Stokes' law predicts that low-density graphite flakes travelling independently should settle slower than denser silicate minerals or large conglomerate grains. Less graphitic CM may be part of a sedimentary conglomerate or the individual particles may be larger and have a higher settling velocity. Thus,





even if Highly Graphitised CM is present as only a minor fraction in the bulk sedimentary input to the ESAS; if other fractions are preferentially deposited on the shelf then the most distal samples will be enriched in Highly Graphitised CM.

The current dataset does not allow definitive determination of whether degradation or distribution is the primary cause of the offshore trend towards Highly Graphitised CM. The Distal ESAS samples are located in deeper water compared to the

other sample groups (see the "Sample Metadata" supplementary table). The deeper water setting allows increased settling time before burial, which would enhance both degradation and winnowing effects. It is noticeable, however, that there are not offshore trends in the distribution of Disordered, Intermediate or Mildly Graphitised CM. Therefore, whichever process is driving the Highly Graphitised CM pattern must be mostly affecting only the crystalline particles.

### 4.2  Principal Component Analysis of spectral groups

In order to observe patterns in the spectral classes between samples and sample groups, principal component analysis (PCA) was applied using the software package R (R Core Team, 2015). For each sample, the proportion of each spectral class was analysed, and principal components derived. Samples were grouped into the classes defined earlier for subsequent analysis. Due to the extreme amounts of Highly Graphitised CM in samples YS-102 and YS-104, the analyses were run with and without these two samples to investigate whether they biased the results.

The values of each principal component are plotted in Figure 5. For this results table, and subsequent discussion, the PCA calculations excluding samples YS-102 and YS-104 are used. PCA results including these two samples are given in Supplementary Figure 3. There was no significant different between the two PCA analysis runs, except that the large amount of Highly Graphitised CM in samples YS-102 and YS-104 created a greater amount of scatter in the Distal ESAS group.

Principal component 1, explaining 74 % of the variance, shows the interplay between Intermediate and Disordered CM.

These two spectral classes plot in opposition to each other. Principal component 2, explaining 22 % of the variance, denotes the relative enrichment in Mildly Graphitised CM. Principal component 3, representing only 4 % of the variance, contains the varying amount of Highly Graphitised CM, but is a minor component and will not be discussed further since the distribution of this material was covered earlier. The various sample groups defined in Table 1 plot in well-defined regions (see Figure 5).

The Kurungnakh Island core plots in the top left corner of Figure 5A. This position is noticeably different to the Buor-Khaya

Bay samples, suggesting that bulk sediment delivered from the Lena River is not the same as permafrost in the downstream area. Since the Lena catchment is one of the largest on Earth, this is not surprising. The nearshore BKB samples are noticeably different to the offshore BKB samples, as shown by their confidence ellipses. Nearshore samples have higher values of PC1 compared to the Kurungnakh Island core (average 0.06 ± 0.12 c.f. -0.5; all errors given as one standard deviation), but show a range of PC2 values ranging from nearly equivalent to substantially lower (average 0.05 ± 0.06, range -0.03 to 0.13, c.f.

0.16). Offshore BKB samples have similar PC1 values to the nearshore samples (average 0.04 ± 0.10), with a lower PC2 value (average -0.11 ± 0.11). This denotes a high amount of Disordered CM, and a low amount of Intermediate and Mildly Graphitised CM, in the Kurungnakh Island core compared to the BKB samples, and that the offshore BKB samples contain more Mildly Graphitised CM than the nearshore BKB samples. As with the Highly Graphtised CM discussed in section 4.1,





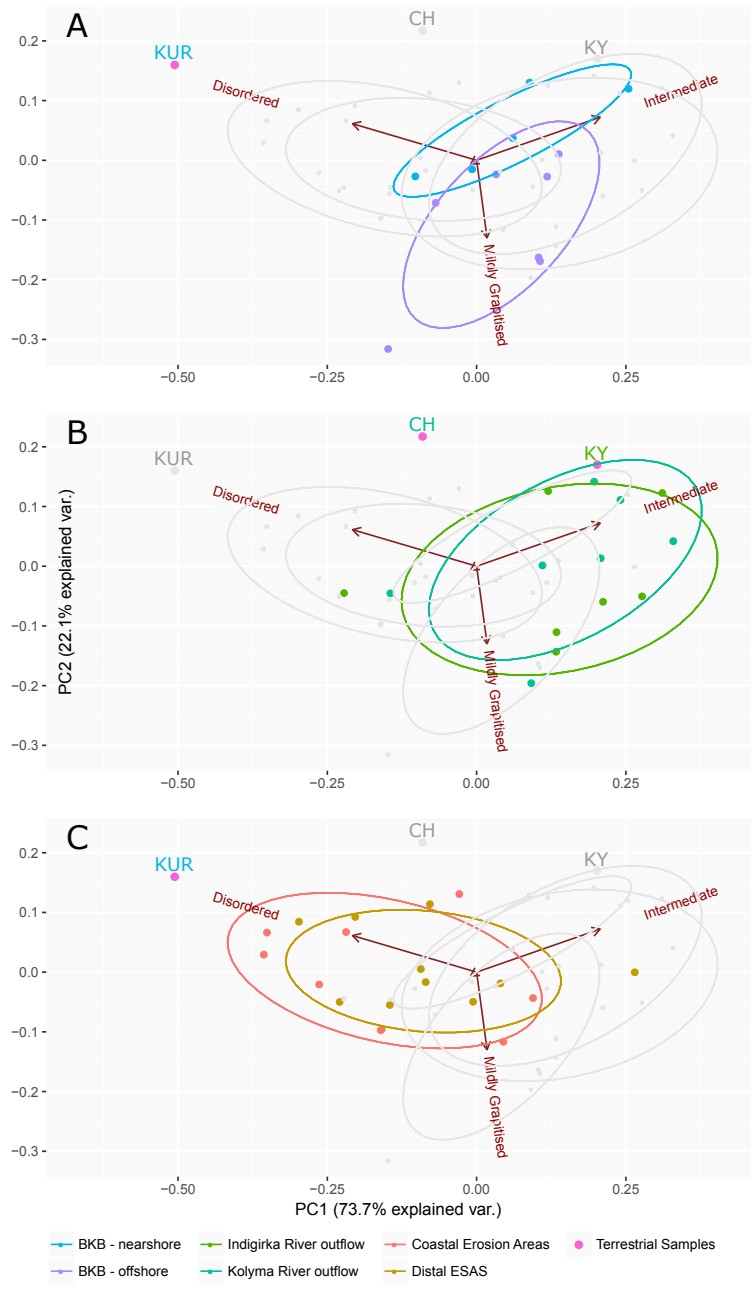

**Figure 5.** Principal Component Analysis plots of PC2 vs PC1 for the sample groups. Terrestrial samples are shown individually. Vectors showing the contribution of spectral classes to the principal components are shown in red. Confidence ellipses (1 standard deviation) are shown in colours matching the sample groups. A: Buor-Khaya Bay nearshore and offshore groups. B: Kolyma and Indigirka river outflows. C: Coastal Erosion and Distal ESAS groups. Data labels showing sample names are included in Supplementary Figure 2.



this could be a degradation or distribution effect, or it could be that the BKB offshore samples have less influence from the Lena outflow and more influence from coastal erosion within the Buor-Khaya Bay, such as Moustakh Island (Vonk et al., 2012).

Onshore permafrost samples from the Indigirka and Kolyma catchments have high values of PC2 (0.171 and 0.218 respectively), which is similar to the Kurungnakh Island core and denotes a low amount of Mildly Graphitised CM. Offshore, the

Indigirka and Kolyma outflows plot in similar locations on Figure 5B. Their respective PC1 values are $0.14 \pm 0.16$ and $0.15 \pm 0.14$; PC2 values are $-0.02 \pm 0.10$ and $0.01 \pm 0.10$. These PC1 values are the highest of all sample groups, denoting that the river outflows are the richest in Intermediate CM and poorest in Disordered CM of the whole ESAS dataset. As seen in the Bour-Khaya Bay, the offshore Indigirka and Kolyma samples have lower PC2 values than the corresponding terrestrial samples, suggesting a higher amount of Mildly Graphitised CM offshore.

The Coastal Erosion dataset has relatively low values of PC1 (average $-0.15 \pm 0.16$) and PC2 values ($-0.00 \pm 0.08$) are similar to the Kolyma, Indigirka, Distal ESAS, and nearshore BKB groups. The low PC1 values denote a high amount of Disordered CM in this group, which is supported by the trend within the sample set as a whole – the confidence ellipse in Figure 5C for this group lies along the vector for Disordered CM.

Samples from the Distal ESAS have similar principal component values to the Coastal Erosion samples. PC1 averages $-0.08$

$\pm 0.15$; PC2 $0.00 \pm 0.07$. This set of values denotes that, similar to the Coastal Erosion group, the Distal ESAS sediments contain high proportions of Disordered CM, low proportions of Intermediate CM, and moderate amounts of Mildly Graphitised CM (more than the terrestrial samples, but less than the offshore Buor-Khaya Bay). It is noticeable that the Distal ESAS confidence ellipse in Figure 5C almost overlays the Coastal Erosion ellipse, and is noticeable different to the ellipses from river outflow groups (Buor-Khaya Bay, Indigirka, Kolyma). This suggests that the Distal ESAS sediments are mostly sourced

from coastal erosion, and that on-shelf homogenisation processes are not sufficient to mix coastal erosion and river derived sediments. It is, however, noticeable that samples YS-102, YS-104 and YS-106 are different to the other Distal ESAS samples, looking similar to the Indigirka Outflow samples, which they correspond to longitudinally.

## 4.3 Spatial pattern of Principal Component 1

To investigate further, the value principal component 1 was plotted across the ESAS (Figure 6). It shows that the Dmitry Laptev

Strait, the region between the Indigirka and Kolyma rivers, and the north eastern part of the distal ESAS, have the lowest values of PC1. Highest values of PC1 are in the outflows of the Indigirka and Kolyma rivers, and in the central distal part of the ESAS (around 150 °E).

In the area offshore the Lena River, the variation is more moderate, trending from mildly positive (somewhat enriched in Intermediate OC) near the river to mildly negative (somewhat enriched in Disordered OC) on the ESAS at 130 °E. These

patterns demonstrate that there are two processes occurring on the shelf. In addition to the general trend towards Highly Graphitised CM that can be seen across the entire ESAS, there are longitudinal zones of Disordered and Intermediate CM. In the western section, Disordered CM is prevalent offshore in contrast to the Buor-Khaya Bay sediments. The permafrost core from Kurungnakh Island is enriched in Disordered CM, as are the samples from the Dmitry Laptev Strait, which is known to





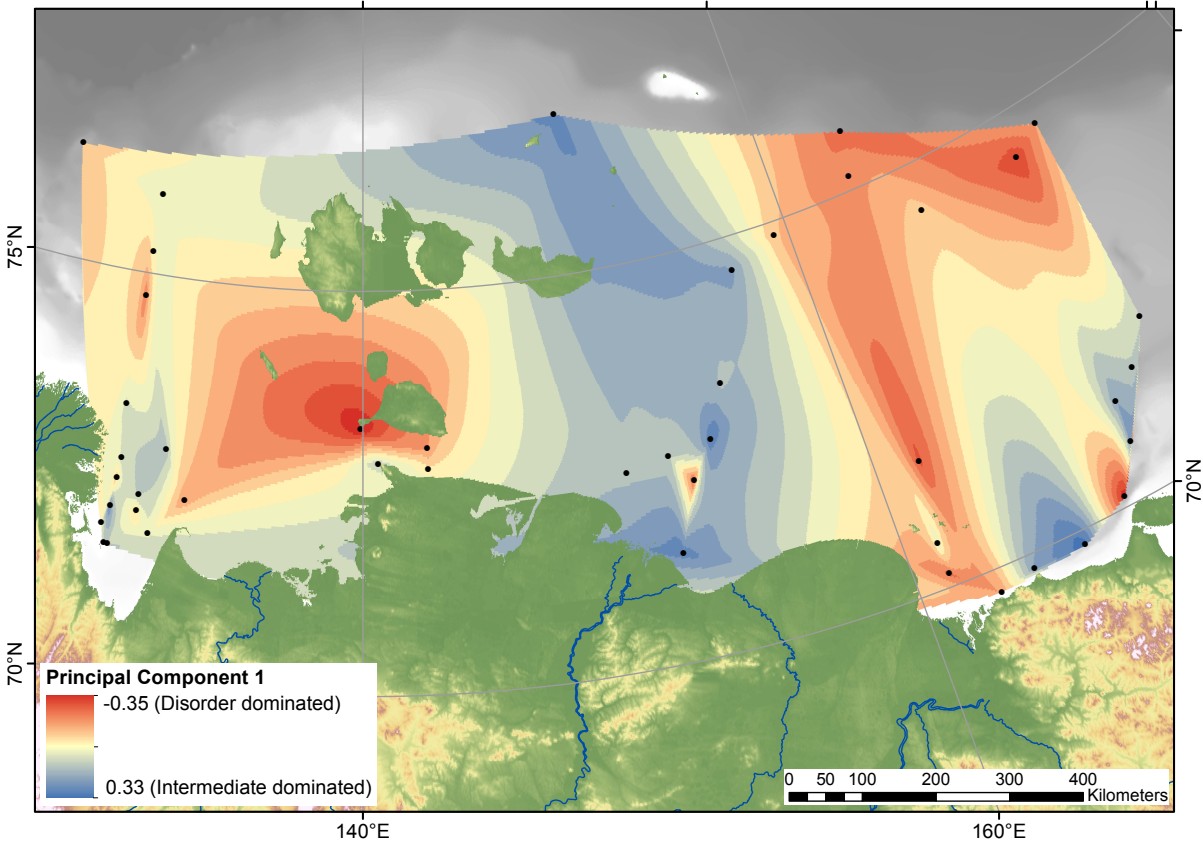

**Figure 6.** Map showing the spatial distribution of principal component 1 across the ESAS. Red colours denote low values of PC1, associated with Disordered CM. Blue colours denote high values of PC1, associated with Intermediate CM.

have high rates of coastal erosion. This suggests that distal sediment in this region may be dominated by coastal erosion of permafrost, and that sediment from the Lena River is less likely to be deposited in the area north of the Lena Delta.

In the central section, from 140 °E to 160 °E, PC1 has a positive value on the Distal ESAS; the proportion of Disordered CM is low, while Intermediate CM dominates. This distribution is similar to that of the Indigirka River outflow at 150 °E. It is

5    likely that the sediments from the Indigirka have been transported out to this part of the shelf. However, there is a possibility that the New Siberian Islands, which are much closer to the Distal ESAS than the Indigirka Outflow, could be the source of this Intermediate CM. Unfortunately there are no samples from the New Siberian Islands for comparison.

In the eastern part of the distal ESAS, beyond 160 °E, PC1 has a negative value that is similar to samples collected to the west of the Kolyma River outflow, but different to the samples collected east of the outflow. Pyrolysis-GC-MS studies of

10    macromolecular OC in this region suggest that most of the OC delivered by the Kolyma River is distributed to the east of the outflow, and that the samples to the west of the outflow are likely dominated by coastal erosion (Sparkes et al., 2016). This would imply that the Distal ESAS sediments east of 160 °E are mostly composed of material eroded from the coastline between



the Indigirka and Kolyma rivers, known to have high rates of coastal erosion (Lantuit et al., 2011). Material delivered by the Kolyma River may be travelling even further to the east, and future sample campaigns beyond 170 °E may find material that is more enriched in Intermediate CM.

### 4.4 Comparison with Soot Black Carbon

Soot black carbon (SBC) is a complementary portion of the recalcitrant OC load in the ESAS. Salvadó et al. (2017) studied SBC using chemical methods (chemothermal oxidation to remove non-SBC, followed by elemental analysis and stable/radiocarbon isotope mass spectrometry), which allowed for quantification of 'SBC' and source apportionment. This method does not directly measure the nature of SBC particles, in contrast to Raman analysis, but allows for quatification rather than investigating relative changes. They showed that whilst the total amount of SBC diminished offshore, along with the proportion of SBC

as a fraction of Total OC, the proportion of SBC as a fraction of permafrost-sourced OC increased offshore, especially in the eastern ESAS. This pattern matches our findings regarding the proportion of Highly Graphitised CM in the ESAS sediments (see Figures 3 and 4). The authors concluded that SBC is recalcitrant compared to other forms of permafrost OC, and avoids degradation during transport. Our Raman study supports this finding, whilst also adding information about the longitudinal distribution of CM. Longidutinal variations in SBC amount, proportion, and radiocarbon signature are small, but Raman suggests

that river-sourced and coastal erosion-sourced recalcitrant organic matter can be differentiated and tracked across the shelf.

One consideration when comparing the chemical SBC and spectroscopic Raman studies is whether there is cross-over between the two carbon pools. SBC particles can have a wide range of chemical structures and grain sizes. Some SBC grains may be large enough to be seen using a Raman microscope (greater than approx. 5 $\mu$m), but most SBC particles are smaller of submicron size and would not be analysed. Some CM, especially Highly Graphitised CM, may survive the chemothermal

oxidation process and be counted amongst the SBC pool. Intermediate and Disordered grade CM is more likely to be lost during the chemothermal oxidation, and it is these two forms of CM that demonstrated longitudinal variation, highlighting the difference between river and coastal permafrost carbon. Thus there is value in measuring both the chemical properties and crystallographic form of recalcitrant organic matter as a way of tracking exported carbon and sediment.

### 4.5 Implications for carbon cycling in the ESAS

The results of this study indicate that terrestrially sourced CM in the East Siberian Arctic region can be found in sediments hundreds of kilometres offshore. Whilst direct quantification of CM as a fraction of total OC is not possible using Raman spectroscopy, chemical analyses show that SBC could account for 14 % of terrestial OC in distal sediments (Salvadó et al., 2017). Highly Graphitised CM is structurally similar to SBC and the chemothermal treatment used in these studies likely removes low-grade CM, implying that CM concentrations could be similar to, if not higher than, SBC offshore.

Whereas biomarker studies suggest that terrestrially sourced organic compounds are often absent or have very low concentrations in offshore settings in this region (Sparkes et al., 2015; Bischoff et al., 2016; Doğrul Selver et al., 2015; Bröder et al., 2016), CM was observed in every offshore sample and there does not seem to be a consistent decline in any of the CM groups – there are Distal ESAS samples with significant proportions of each of these classes. Halfway between extractable





lipids and Raman-amenable CM particles is macromolecular organic matter, such as lignin. Tesi et al. (2014) extracted these same samples and found that concentrations of lignin phenols were also very low in the outer shelf. Thus the CM observed using Raman behaves very differently to extractable and pyrolysable OC. While there are observable offshore trends, it appears that this material is much more resilient than the labile organic matter traditionally used to track OC export from terrestrial

settings. This observation likely accounts for the "old" radiocarbon values observed in ESAS sediments ((Sparkes et al., 2016; Vonk et al., 2012), where carbon eroded from coastal sediments is thought to be major contributor to offshore sedimentary OC, whereas modern terrestrial OC is a minor component far offshore. CM has undergone diagenesis and/or metamorphism, a process that takes a significant amount of time, and should therefore have no radiocarbon remaining. Conservative offshore transport also addresses the anomalous radiocarbon values measured in the Beaufort Sea by Goñi et al. (2005). If CM can be

transported for large distances offshore without significant degradation, it could lead to unusually "old" radiocarbon ages in seafloor sediments, but potentially also suspended particulate matter.

   Long-distance CM transport also has implications for the carbon cycle in this region. Vonk et al. (2012) estimated, using bulk isotope analysis, that 66 % ± 16 % of the 44 ± 10 Tg OC eroded from ice complexes along the ESAS coastline is released as $CO_2$, with the remaining third buried on the shelf or deep ocean. Distal shelf sediments were estimated as containing up to

50 % of their OC from coastal erosion sources. However, subsequent molecular analyses have suggested that organic matter degradation is extremely prevalent by the outer shelf. Solvent extractable material, while only accounting for 5 to 10 % of the total OC content of exported sediment, have predicted a significant loss of terrestrial OC during transport across the ESAS (Bischoff et al., 2016; Bröder et al., 2016; Karlsson et al., 2015; Sparkes et al., 2015; Tesi et al., 2014). Pyrolysis GC-MS, investigating larger, non-extracted, biomolecules also found that terrestrial markers were much diminished on the outer shelf

(Sparkes et al., 2016). Therefore, this study provides a systematic characterisation of the most likely form of terrestrial carbon in distal offshore settings. Erosion of recalcitrant OC from permafrost, without subsequent degradation, will transpose carbon from land into ESAS sediments or the deep Arctic Ocean basin, with no net effect on atmospheric $CO_2$ levels. If the recalcitrant CM analysed in this study forms a significant proportion of the OC load of the ESAS, then the warming-induced carbon cycle feedbacks will not be as severe as if all terrestrial OC was degraded. Thus, when modelling the impact of permafrost erosion on

climate change, and vice versa, researchers should consider not the Total Organic Carbon content of the mobilised sediment, but the labile fraction of this. Note that some of this labile fraction could be ancient carbon, currently locked in permafrost deposits across eastern Siberia (Tarnocai et al., 2009), and therefore permafrost thaw can still cause a net increase in atmospheric $CO_2$ levels, and an associated positive feedback relationship with climate warming.

## 5   Conclusions

Raman Spectroscopy of Carbonaceous Material has been successfully applied across the East Siberian Arctic Shelf, a complex and heterogeneous sedimentary system. By grouping collected spectra into classes, and applying principal component analysis, CM from river and coastal erosion has been differentiated – coastal erosion delivers more Disordered CM, while rivers deliver more Intermediate CM. Across the ESAS, two main processes have been identified. Firstly, there is a trend across the entire





shelf towards Highly Graphitised CM in the furthest offshore samples, those collected from the Distal ESAS. This could be due to degradation of weaker CM during transport, or preferential winnowing of crystalline graphite to more distal settings; given the presence of Disordered CM in the most distal settings, winnowing is the preferred explanation. In addition to this, an East-West pattern is observed in the distribution of Disordered ad Intermediate CM. This suggests that there are areas

where sedimentation is dominated by material sourced from rivers, and areas dominated by coastal erosion sediment. Despite the shelf being extremely wide and shallow, sediment transport processes have not homogenised the sediments and three individual sections can be identified along the shelf. Raman Spectroscopy is thus a valuable tool for identifying and tracking eroded sediment in complex environments. Furthermore, the presence of CM ranging from Disordered to Highly Graphitised in distal sediments, hundreds of kilometres from the coastline, shows that Raman-amenable carbon is generally resistant to

oxidation and thus a conservative part of the organic carbon cycle in this region.

*Code and data availability.*   Supplementary information is supplied with this paper. Supporting data and code underlying this study are available via the MMU research repository at http://researchdata.mmu.ac.uk/id/eprint/76

Supplementary information includes:

– Sample Metadata: A spreadsheet containing sample locations and previously measured geochemical parameters

– Supplementary Figure 1: An example of peak fitting results for Disordered/Intermediate, Mildly Graphitised and Highly Graphitised spectra

– Supplementary Figure 2: Principal Component Analysis results, as shown in Figure 5, but with each sample location identified

– Supplementary Figure 3: Principal Component Analysis results if samples YS-102 and YS-104 are also included in the procedure

Supporting data and code include:

– Unprocessed Raman spectra as x,y text files,

– Fitted spectra, showing the automatically fitted peaks, as a PDF file

– Parameters of the fitted peaks including height, width, location and area

– The shell script used to fit Raman data

Raman fitting procedures are also published and maintained at https://github.com/robertsparkes/raman-fitting.

*Author contributions.*   Ö. Gustafsson, B. E. van Dongen, and I. P. Semiletov collected samples along with the crew of ISSS-08. R. B. Sparkes designed the study. A. Dogrul Selver assisted in preparing the samples for analysis. Raman Spectroscopy measurements were carried out by M. Maher, J. Blewett and R. B. Sparkes. Data analysis was carried out by R.B. Sparkes. R. B. Sparkes and B. E. van Dongen prepared the manuscript with contributions from all co-authors.

*Competing interests.*   The authors declare that they have no conflict of interest.





*Acknowledgements.* We gratefully acknowledge receipt of a NERC research grant (NE/I024798/1) to B. E. van Dongen, an MMU studentship to R. B. Sparkes/M. Maher, an MMU-EERC research grant to R. B. Sparkes, Swedish Research Council (VR Contracts 621-2007-4631, 621-2013-5297 and the Distinguished Professors Grant 2017-01601) and the European Research Council (ERC-AdG CC-TOP project #695331) support to O.Gustafsson, and support from the Government of the Russian Federation (mega-grant 14.Z50.31.0012) to I. Semiletov. We

5   thank the crew and personnel of the R/V Yakob Smirnitskyi and all colleagues in the International Siberian Shelf Study (ISSS) Program for support, including sampling. We thank T. Tesi for providing the Yedoma samples for the Kolyma and Indigirka catchment areas. We thank J. Bischoff and D. Wagner for providing the Kurungnakh Island core. H. Andrews provided technical support for the MMU Raman Spectrometer. The ISSS program is supported by the Knut and Alice Wallenberg Foundation, the Far Eastern Branch of the Russian Academy of Sciences, the Swedish Research Council (VR Contract No. 621-2004-4039, 621-2007-4631 and 621-2013-5297), the US National Oceanic

10   and Atmospheric Administration (OAR Climate Program Office, NA08OAR4600758/Siberian Shelf Study), the Russian Foundation of Basic Research (08-05-13572, 08-05-00191-a, and 07-05-00050a), the Swedish Polar Research Secretariat, the Nordic Council of Ministers and the US National Science Foundation (OPP ARC 0909546). %Finally, we thank the associate editor and XXX





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
