# Peer review of "Carbonaceous material export from Siberian permafrost tracked across the Arctic Shelf using Raman spectroscopy"

_The Cryosphere, 2018_

## Referee Comment (RC1) · Anonymous Referee #1 · 31 Mar 2018

This paper uses Raman spectroscopy to investigate the composition and source of carbonaceous material (CM) exported into the East Siberian Arctic Shelf (ESAS) in comparison with those in terrestrial and coastal erosion samples. Based on peak characteristics using deconvolution techniques, they classified CM into Disordered, Intermediate, Mildly Graphitised and Highly Graphitised groups and observed an enrichment of Intermediate CM in sediments exported from the Indigirka and Kolyma rivers versus Highly Graphitised CM in distal samples. As Raman spectroscopy examines a slightly different pool of carbon materials (and properties) as those previously analyzed by chemothermal oxidation on the ESAS sediments, this paper adds complementary information on the fate of terrestrially derived ancient (graphite like) organic matter in

the Arctic Ocean. As these materials are considered to be highly recalcitrant derived from fossil (rock) carbon and very old, it is important to tease them apart from biologically synthesized, permafrost OC to precisely assess the fate of permafrost carbon in the Arctic. I think the dataset is unique and original that deserves publication to improve our understanding of the Arctic carbon cycling.

That being said, I have a few questions and/or suggestions on the statistics and techniques used in the paper.

1. The main conclusion of the paper is based on the comparison of CM in samples from different locations. While a quite large number of Raman spectra were collected, there is no description of how statistical analysis was carried out in the Methods section—for instance, how were differences determined for datasets with different number of samples? What statistical analysis was used? How was error propagated for datasets with analytical errors and replicates? How was PCA carried out? Were the data normally distributed? These are key questions regarding the robustness of the conclusion.

Actually, looking at Table 2, I would think that the statistical differences only occur for the Highly Graphitised CM in distal ESAS samples relative to the others and for Mildly Graphitised CM in terrestrial samples relative to the others. The others are mostly similar with a big standard deviation. I recommend using box graphs showing the median, mean and percentiles for each sample group rather than using Figure 3 (which is redundant showing less information than Table 2).

2. Regarding the presentation of data, I prefer to see the distribution of CM drawn in the format of Figure 2 rather than in Figures 4 and 6. While the latter is truly impressive, how reliable are the schemes given the scattered and uneven distribution of sampling locations?

3. The authors mentioned that black carbon particles smaller than submicron size is not detectable by Raman spectroscopy. However, it is also mentioned in the methods

that hours of grinding does not affect CM crystallinity, suggesting no effects on Raman spectroscopy. I am a bit confused here. How big is the pool of black carbon "undetectable" for Raman in the total CM or black carbon budget? Is it possible that, during transport and winnowing, CM may be physically ground to smaller particles to escape the analytical window? How would this affect your data interpretation? In the end, I think it is very important to frankly point out drawbacks of the method as no method is perfect.

4. For the discussion part, I think it makes more sense to introduce PCA analysis first, followed by comparison of group mean values. I also think that some descriptions are repeated and can be shortened to increase the readability.

There are some minor mistakes: Page 13: Line 15: . . .have been caused by. . . Line 29: . . . is that it is preferentially. . . Page 14: Line 7: no offshore trends

---

## Referee Comment (RC2) · Anonymous Referee #2 · 11 Apr 2018

General comments:

This paper applied Raman Spectroscopy to track carbonaceous material export from Siberian permafrost region. This is the first application of Raman Spectroscopy on sedimentary CM in the Arctic which showed very interesting findings. In this study, samples were mostly marine surface sediments from nearshore to distal shelf of three major Arctic rivers in East Siberia and few terrestrial permafrost samples. Samples was classified into four class based on Raman peak areas and widths. After statistical analysis of all the data, they found that highly graphitized CM was higher in the distal slope of than the rest of the shelf which is more likely a result of winnowing effect. In addition,

they found that there is longitudinal variation of sources of CM to the deeper shelf from riverine inputs and coastal erosion. This information highlighted the importance of CM in carbon cycling in the Arctic, especially for predicting future climate, since this recalcitrant part of total OC may not have effect on atmospheric $CO_2$ levels. In addition, their work introduced another technique for tracking permafrost mobilization in long distance in the Arctic, apart from bulk and biomarker which experience severe degradation during transport. The content of this paper is about CM export from permafrost to the Arctic Ocean, so it is under the general scope of the Cryosphere. This paper is well-organized and clearly stated. Large sample size, wide distribution and multi-sampling of each sample were sufficient to support their interpretations and conclusions. The experiment and concepts are well described. The author properly cited related paper and showed the specialty of this study. As a result, I recommend the publication of this paper in The Cyrosphere. There are still some minor issues as list under specific comments.

Specific comments:

Page 5 line 5. What is the reference for "over decadal to millennial timescales...into account.

Page 5 line 21-22. "Three terrestrial location ...catchment" This sentence needs to be modified because the three sites can not represent what is coming from the catchment. Even though this is clarified later, it should be stated here as well.

Page 5 line 23-25. Each terrestrial location has three samples analyzed. How different are they on those parameter? Only averages are shown in the supplementary data. Vertical difference could be very significant in permafrost cores.

Page 6 Figure 1. It would be nice to have ICD distribution on this map.

Page 8 Table 1. In the table statement, Tmin and Tmax were not mentioned. Even though T is related to R2 and RA2 ratio, it is still better to keep consistence in statement

and table content.

Page 9 section 3.2. List of grouping is not in the sample order as the figure 2. It would make more sense to keep the same order.

Page 11. Table 2. I think it would be better to put the three terrestrial sites separately so that it would be easier to compare with individual river outflow.

Page 16 line 19-20. "are mostly sourced from coastal erosion". This may be true. But can you give some grain-size evidence to better support this? What is the difference between coastal inputs and riverine inputs? Which one has higher amount of fine particles?

Line 21. "noticeable that YS-102…distal ESASsamples". Please highlight those dots if you want to talk about them. I did not see three dots that are distinguishingly different from others.

Technical corretions: Page 2, line5. Period is missing prior to Deepening. Page 3, line 25. Superscipt for -1 Line 29. Delete "and" after power. Page 5 line 5. Add "than" after "This is a much larger system". Figure 5 b: The two colors are too hard to distinguish.

---

## Author Comment (AC1) · 23 Jun 2018

**Response to Reviewer 1**

We thank the reviewer for their positive comments, and hope that these responses are satisfactory

**Comment:** The main conclusion of the paper is based on the comparison of CM in samples from different locations. While a quite large number of Raman spectra were collected, there is no description of how statistical analysis was carried out in the Methods section, for instance, how were differences determined for datasets with different number of samples? What statistical analysis was used? How was error propagated for datasets with analytical errors and replicates? How was PCA carried out? Were the data normally distributed? These are key questions regarding the robustness of the conclusion.

Actually, looking at Table 2, I would think that the statistical differences only occur for the Highly Graphitised CM in distal ESAS samples relative to the others and for Mildly Graphitised CM in terrestrial samples relative to the others. The others are mostly similar with a big standard deviation. I recommend using box graphs showing the median, mean and percentiles for each sample group rather than using Figure 3 (which is redundant showing less information than Table 2).

> **Response:** We agree that statistical analyses form a key portion of the manuscript, and will make suitable changes to the text to explain these to the reader. Briefly, significance was calculated using the mean, standard deviation and number of samples, using a t test calculation to generate P values. PCA was carried out using the 'prcomp' function within the software package R. Instrumental and sampling error were not investigated for this study. Instrumental procedures were previously tested in detail (Sparkes et al., 2013). Each sample consisted of spectra collected from ~30 individual pieces of organic carbon, and a repeated sampling could have led to a different subset of organic particles being measured. However, the consistency within and demonstrable difference between each group of samples, is in line with previous analyses, indicating that 30 spectra per sample is enough to produce a robust characterisation of each location.
>
> Offshore Raman studies involve the mixing together of terrestrial OC from multiple sources, and so the null hypothesis in any such project should be a homogenous distribution across the shelf. Figures 2 and 3, and Table 2, demonstrate this trend – there is a large degree of similarity between the samples, but careful PCA analysis shows that small differences seen in the distribution of the different spectral classes are significant and systematic.
>
> *We will produce the box graphs requested by the reviewer and include them in place of Figure 3 in the revised manuscript.*

**Comment:** Regarding the presentation of data, I prefer to see the distribution of CM drawn in the format of Figure 2 rather than in Figures 4 and 6. While the latter is truly impressive, how reliable are the schemes given the scattered and uneven distribution of sampling locations?

**Response:** This presentation style, using standard algorithms from ArcMap, has been employed in a number of studies, utilising the ISSS-08 samples and around the world (see examples from the reference list: Vonk et al., 2012; Sparkes et al., 2015; Bischoff et al., 2016; Sparkes et al., 2016). Non-uniform sample distributions are never ideal but we feel that there are enough sample locations to support these interpolations.

*We propose including the requested figures as supplementary information, if required by the editor.*

**Comment:** The authors mentioned that black carbon particles smaller than submicron size is not detectable by Raman spectroscopy. However, it is also mentioned in the methods that hours of grinding does not affect CM crystallinity, suggesting no effects on Raman spectroscopy. I am a bit confused here. How big is the pool of black carbon "undetectable" for Raman in the total CM or black carbon budget? Is it possible that, during transport and winnowing, CM may be physically ground to smaller particles to escape the analytical window? How would this affect your data interpretation? In the end, I think it is very important to frankly point out drawbacks of the method as no method is perfect.

**Response:** Grinding for many hours reduces grain size but does not affect the degree of crystallinity of the graphite particles (see Nakamizo et al., 1978; Sparkes et al., 2013). Therefore, there is a risk of material becoming too small for the analytical window, but until that point the technique should robustly characterise the distribution of CM types.

Using a 50x objective, as this study employed, Raman spectroscopy can measure particles of about 1-2 μm and above. This includes silt and clay size fractions, and even in distal samples consisting of fine mud, individual grains can still be identified clearly. Colloidal material smaller than this is less likely to settle in shelf sediments, and could be winnowed into the deep ocean. The fraction of material ground to this small size is hard to quantify, but we believe that the majority of the CM present is within the appropriate size window for Raman analysis. Whether the material is sufficiently crystalline to produce measurable Raman spectra is another matter. Raman analysis of atmospheric SBC produces spectra with very broad peaks (Catelani et al., 2014), similar to very disordered lignite-grade terrestrial CM.

Atmospheric soot particles are typically in the nm range, these would be too small for this study and would not be identified as individual CM particles. Large SBC particles, eroded and transported by fluvial and coastal processes, alongside sedimentary matter, may be included in this analysis.

*We are happy to clarify all of this in the text.*

**Comment:** For the discussion part, I think it makes more sense to introduce PCA analysis first, followed by comparison of group mean values. I also think that some descriptions are repeated and can be shortened to increase the readability.

**Response:** *We are happy to switch sections 4.1, 4.2 and 4.3 so that PCA analysis is introduced and discussed before Highly Graphitised CM. Suitable alterations will be*

*made to ensure continued internal consistency, and reduce redundancy as identified by the reviewer.*

**Comment:** There are some minor mistakes:
Page 13: Line 15: *…have been caused by…*
Line 29: *… is that it is preferentially…*
Page 14: Line 7: no offshore trends

**Response:** *We will change the text accordingly.*

---

## Author Comment (AC2) · 23 Jun 2018

**Response to Reviewer 2**

We thank the reviewer for their positive and constructive comments, and hope that the amendments outlined below address them

**Comment:** Page 5 line 5. What is the reference for "over decadal to millennial timescales...into account.

  **Response:** *We will remove this statement since it is not relevant to the manuscript*

**Comment:** Page 5 line 21-22. "Three terrestrial location ...catchment" This sentence needs to be modified because the three sites can not represent what is coming from the catchment. Even though this is clarified later, it should be stated here as well.

  **Response:** *We will make this clarification, thank you for bringing it to our attention*

**Comment:** Page 5 line 23-25. Each terrestrial location has three samples analyzed. How different are they on those parameter? Only averages are shown in the supplementary data. Vertical difference could be very significant in permafrost cores.

  **Response:** At each terrestrial location, the three samples were very similar.

  *We will make sure that this is clear in the manuscript or data tables.*

**Comment:** Page 6 Figure 1. It would be nice to have ICD distribution on this map.

  **Response:** We do not have access to a suitable dataset to show ICD distribution. As a proxy for this, areas of high coastal erosion are likely to be ICD areas, since the weakness of ICD contributes to rapid erosion rates.

**Comment:** Page 8 Table 1. In the table statement, Tmin and Tmax were not mentioned. Even though T is related to R2 and RA2 ratio, it is still better to keep consistence in statement and table content.

  **Response:** *We will alter the table caption to read "Parameters for classifying Raman spectra into four groups based on their metamorphic temperature (determined from the R2 or RA2 peak area ratio; see Sparkes et al., 2013), and Total width parameter (G+ D1 + D2).*

**Comment:** Page 9 section 3.2. List of grouping is not in the sample order as the figure 2. It would make more sense to keep the same order.

  **Response:** *We will make sure the list is in the same order as in Figure 1, and update the in-text reference to refer to the correct figure.*

**Comment:** Page 11. Table 2. I think it would be better to put the three terrestrial sites separately so that it would be easier to compare with individual river outflow.

  **Response:** *We will adjust the table and include the three sites separately.*

**Comment:** Page 16 line 19-20. "are mostly sourced from coastal erosion". This may be true. But can you give some grain-size evidence to better support this? What is the difference between coastal inputs and riverine inputs? Which one has higher amount of fine particles?

> **Response:** Geochemical studies have shown that distal sediments are dominated by coastal erosion material (see Sparkes et al., 2016, Bischoff et al., 2016, Vonk et al., 2012). Grain size distributions, coupled with organic analyses, have been measured by Tesi et al., 2016, and
>
> *We will reference Tesi et al., 2016, to support our assertion.*

**Comment:** Line 21. "noticeable that YS-102...distal ESASsamples". Please highlight those dots if you want to talk about them. I did not see three dots that are distinguishingly different from others.

> **Response:** *We will remove this observation, it is not relevant to the discussion in this paragraph.*

**Comment:** Technical corrections:
Page 2, line5. Period is missing prior to Deepening.
Page 3, line 25. Superscipt for -1 Line 29. Delete "and" after power.
Page 5 line 5. Add "than" after "This is a much larger system".
Figure 5 b: The two colors are too hard to distinguish.

> **Response:** *We will change the text accordingly*